# Genetic Basis of Resistance to *Warrior (-)* Yellow Rust Race at the Seedling Stage in Current Central and Northern European Winter Wheat Germplasm

**DOI:** 10.3390/plants12030420

**Published:** 2023-01-17

**Authors:** Fahimeh Shahinnia, Volker Mohler, Lorenz Hartl

**Affiliations:** Bavarian State Research Center for Agriculture, Institute for Crop Science and Plant Breeding, 85354 Freising, Germany

**Keywords:** *Warrior (-)* race, stripe rust, GWAS, *Yr29/Lr46* gene, European wheat, SNP marker

## Abstract

To evaluate genetic variability and seedling plant response to a dominating *Warrior (-)* race of yellow rust in Northern and Central European germplasm, we used a population of 229 winter wheat cultivars and breeding lines for a genome-wide association study (GWAS). A wide variation in yellow rust disease severity (based on infection types 1–9) was observed in this panel. Four breeding lines, TS049 (from Austria), TS111, TS185, and TS229 (from Germany), and one cultivar, TS158 (KWS Talent), from Germany were found to be resistant to *Warrior (-) FS 53/20* and *Warrior (-) G 23/19*. The GWAS identified five significant SNPs associated with yellow rust on chromosomes 1B, 2A, 5B, and 7A for *Warrior (-) FS 53/20*, while one SNP on chromosome 5B was associated with disease for *Warrior (-) G 23/19*. For *Warrior (-) FS 53/20*, we discovered a new QTL for yellow rust resistance associated with the marker *Kukri_c5357_323* on chromosome 1B. The resistant alleles G and T at the marker loci *Kukri_c5357_323* on chromosome 1B and *Excalibur_c17489_804* on chromosome 5B showed the largest effects (1.21 and 0.81, respectively) on the severity of *Warrior (-) FS 53/20* and *Warrior (-) G 23/19*. Our results provide the basis for knowledge-based resistance breeding in the face of the enormous impact of the *Warrior (-)* race on wheat production in Europe.

## 1. Introduction

In 1894, Eriksson and Henning recognized yellow rust (also known as stripe rust) as a distinct rust disease in which the pathogen *Puccinia striiformis* Westend. (*Pst*) can attack wheat, rye, barley, and 59 grass species [1,2]. The fungal pathogen produces yellow to orange uredinia mainly on leaf blades, but also on leaf sheaths, stems, glumes, awns, and young grains of susceptible plants. When leaves are covered by uredinia, photosynthesis is severely limited and continuous production of urediniospors deprives host plants of water and nutrients, reducing plant growth, the number of ears and grains per ear, and test weight. The disease can result in an average loss of 13% of grain yield [3] and up to 100% in fields planted with highly susceptible cultivars under extreme yellow-rust-friendly weather conditions [4,5]. Chlorosis or necrosis (hypersensitive reaction) is a disease symptom in resistant plants [6]. The intensity of the reaction depends on the degree of plant resistance and the three main weather conditions, including wind, moisture, and temperature. Yellow rust has been considered primarily a disease of cooler climates (2–15 °C) and higher and northern elevations, but recent epidemics of the disease indicate that fresh strains show greater adaptation to higher temperatures and countries near the equator [2].

Genetic resistance to *Pst* in wheat is based on the effect of genes (major, minor), number of genes (monogenic, polygenic), inheritance of genes (qualitative, quantitative), and molecular basis of genes (NBS-LRR-type resistance, non-NBS-LRR-type resistance) [7]. However, depending on race specificity, growth stage, and temperature sensitivity, types of resistance to yellow rust can be divided into race-specific resistance for all growth stages and non-race-specific resistance for the adult plant stage [7]. The evolution of new races of *Pst* through mutation and somatic and sexual recombination results in a significant change in the virulence of the pathogen, making it more capable of overcoming genetic resistance and plant defence mechanisms.

The most economical and effective method of controlling yellow rust is genetic resistance, combining both minor and major resistance genes. Breeding for rust resistance has used both race-specific and race-nonspecific or partial resistance genes [4]. However, high genetic variation in the pathogen population and rapid selection of new virulent races have forced breeders to focus on pyramiding strategies that combine multiple race-specific and/or race-nonspecific resistance genes to increase the durability of resistant cultivars [8]. Therefore, identification of sources of effective resistance genes and their molecular characterization is an ongoing process to ensure genetic diversity in breeding programs. The main objective is therefore to develop advanced lines or cultivars with high and stable yield potential that also have durable rust resistance.

In 2011, “*Warrior”*, a new virulent yellow rust strain from a region near the Himalayas [9], appeared simultaneously in several European countries and spread rapidly across much of the continent. According to observations by the Julius Kühn Institute (JKI, Germany), the “*Warrior (-)*” race, which belongs to the *PstS10* group, dominates the European yellow rust population [10]. It appears that only a few resistance genes, including *Yr5, Yr10, Yr15*, and *Yr27*, are still effective against these races in Europe (K. Flath 2022, JKI, personal communication). Despite the enormous impact of the *Warrior (-)* race on wheat production in Europe, little is known about the genetic control of resistance to this race, which provides the basis for knowledge-based resistance breeding [11]. Quantitative trait locus (QTL) analysis using classical bi-parental linkage mapping and genome-wide association studies using large diversity panels have successfully identified genomic regions associated with yellow rust resistance and contributed to a better understanding of the genetic basis of disease resistance in both seedling and adult growth stages [8,11,12,13,14,15,16]. In addition, the identification of diagnostic molecular markers associated with yellow rust genes has improved our ability to rapidly incorporate resistance into breeding material through marker-assisted selection (MAS). With the advent of high-throughput sequencing and SNP genotyping techniques on universal bead arrays [17] and the availability of the wheat reference genome [18], the search for candidate genes and development of functional markers for use in MAS has been greatly facilitated.

In this study, we used an association panel of winter wheat consisting of 229 commercial cultivars and advanced breeding lines selected from Central and Northern European wheat breeding programs, previously published by Shahinnia et al. [16] with the exception of line TS 175, since no seed was available to perform the experiments. Our objectives were to (1) conduct an association mapping study using genome-wide SNP markers to identify chromosomal regions associated with *Warrior (-)* yellow rust resistance at the seedling stage, (2) identify sources of effective resistance alleles and associated QTLs for use in breeding programs, and (3) determine the relationship between QTLs identified in this study and known genes and QTLs for yellow rust resistance in previous studies.

## 2. Results

### 2.1. Phenotypic Evaluation for Warrior (-) Yellow Rust

A wide variation was observed in yellow rust disease severity based on infection types (IT) 1–9, ranging from very resistant to very susceptible in the association panel (Figure 1), the details of which are written in the Materials and Methods section.

In the seedling test of lines against the *Pst* pathotype *Warrior (-) FS 53/20* (mean IT scores = 6.15 ± 0.09), 16% of genotypes were resistant and 33% were susceptible, while 51% of genotypes showed moderate resistance. A different trend was observed for the *Pst* pathotype *Warrior (-) G 23/19* (mean IT scores = 6.23 ± 0.10), in which 46% of all genotypes showed different degrees of susceptibility and only 9% showed resistant reaction types, while 45% showed moderate resistance. The mean IT values of the two subpopulations (previously identified by structural analysis [16]) for *Warrior (-) FS 53/20* (6.4 vs. 5.9) and *Warrior (-) G 23/19* (5.8 vs. 6.4) were in the moderately resistant group (IT 5–6) and showed no significant differences (Appendix A). Four breeding lines, TS049 (from Austria), TS111, TS185, and TS229 (from Germany), and one variety, TS158 (KWS Talent) from Germany, were found to be resistant to *Warrior (-) FS 53/20* and *Warrior (-) G 23/19* (Appendix A). The frequency distribution of infection type for resistant and susceptible genotypes based on mean values is shown in Figure 1. ANOVA also revealed significant differences (*p* < 0.01) within the group of genotypes tested against *Warrior (-) FS 53/20* (mean square = 167.83) and *Warrior (-) G 23/19* (mean square = 28.93). The correlations (*r^2^* = 0.39 to 0.54) between replications for the *Warrior (-) FS 53/20* were smaller than the correlations (*r^2^* = 0.78 to 0.82) for *Warrior (-) G 23/19* in our study. A low positive correlation (*r^2^* = 0.38) was observed between the yellow rust responses to the two *Pst* pathotypes recorded based on the mean values of the genotypes.

### 2.2. GWAS for Resistance to Warrior (-) Yellow Rust at Seedling Stage

A GWAS using a mixed linear model identified five significant SNPs associated with yellow rust on chromosomes 1B, 2A, 5B, and 7A for *Warrior (-) FS 53/20*, while one SNP was associated with the disease on chromosome 5B for *Warrior (-) G 23/19*, based on the mean values obtained from winter wheat panel evaluation at the seedling stage (Table 1, Figure 2). The percentage of explained phenotypic variance (*R^2^*) of the associated markers ranged from 5% to 6%, whereas the effect size of these markers ranged from 0.75 to 1.21 (Table 1). The strongest association (*R^2^* = 6%, *p*-value = 0.0001) was found for SNP marker *BS00062869_51* on chromosome 7A for *Warrior (-) FS 53/20* (Table 1). For *Warrior (-) FS 53/20,* the majority of lines (193) carried the resistant allele A at the marker locus *Tdurum_contig13879_352*, while 104 lines carried the resistant allele T at the marker locus *Excalibur_c17489_804* for *Warrior (-) G 23/19*. The resistant allele G at the marker locus *Kukri_c5357_323* on chromosome 1B and the resistant allele T at the marker locus *Excalibur_c17489_804* on chromosome 5B showed the greatest effects on the severity of yellow rust detected for *Warrior (-) FS 53/20* and *Warrior (-) G 23/19*, respectively.

The QQ plots evaluating the performance of the mixed linear models for comparing the observed −Log_10_ (*p*-value) vs. expected −Log_10_ (*p*-value) showed a high fit of the GWAS model (Figure 2).

Lines TS049, TS111, TS185, and TS229, which were found to be resistant to *Warrior (-) FS 53/20* and *Warrior (-) G 23/19*, contained 3–5 resistant alleles (G, A, C, C, A, and T) of significantly associated SNPs identified by the GWAS (Table 2).

To identify stable QTLs, a GWAS was also performed for each experimental replicate per *Pst* pathotype (Appendix A, Appendix A). For *Warrior (-) FS 53/20*, except for the SNP associated with stripe rust resistance at the marker locus *CAP12_c703_150* on chromosome 5B, other SNPs on chromosome 1B (*Kukri_c5357_323* and *Tdurum_contig13879_352*) were also identified in the first and second replicates of the experiment, whereas SNPs on chromosome 2A and 7A (*BS00098033_51* and *BS00062869_51,* respectively) were also detected in the third replicate of the experiments (Appendix A). For *Warrior (-) G 23/19*, the significantly associated marker on chromosome 5B (*Excalibur_c17489_804*) was exclusively identified based on the mean values of the three replicates (Appendix A).

### 2.3. Correspondence to Published Yr Loci

The analysis of physical positions for flanking markers related to the associated SNPs based on *The Catalogue of Gene Symbols for Wheat* led to the identification of five correspondence intervals, *IWA3095-IWB46805*, *IWA5959-wPt-7765*, *IWA8097-IWB57930*, *IWA4528-IWA5336*, and *IWB2716-IWB53860*, previously mapped on chromosomes 1B, 2A, 5B, 7A, and 5B, respectively (Appendix A). 

## 3. Discussion

To expand the spectrum of resistance to yellow rust, the development and use of resistant cultivars with new alleles is a forward-looking way to control this disease economically and ecologically through breeding methods. In the present study, we focused on analysing the response of wheat seedlings to two single spore isolates of *Warrior (-)* yellow rust. The six SNP loci (Table 1) associated with the severity of yellow rust infection (based on the average of infection type in the experiments) represented QTLs with small effects (*R^2^* less than 10%). Such low-effect QTLs for resistance to yellow rust at the seedling stage were also identified in the U.S. elite spring wheat lines [19], the German MAGIC winter wheat population [20], and a Chinese wheat landrace association panel [5]. About 50% of the lines in our study showed moderate resistance to both *Warrior (-)* races at the seedling stage, indicating the presence of a relatively large number of genes with a moderate effect. About 10% of the lines showed a high level of resistance, which could be caused by major resistance genes against both pathotypes or by a combination of a few resistance genes. However, these major resistance genes remained undetected in the GWAS, possibly due to low allele frequencies. According to the Global Rust Reference Center (https://agro.au.dk/forskning/internationale-platforme/wheatrust/yellow-rust-tools-maps-and-charts/races-changes-across-years, accessed on 12 October 2022), *Warrior (-)* was the dominant yellow rust pathogen in European countries from 2014 to 2019, apart from the *Amboise* race in 2020-2021, both of which belong to the *PstS10* group. According to the results of the RustWatch project (European Early Warning System for Wheat Rust Disease, https://ec.europa.eu/eip/agriculture/en/find-connect/projects/rustwatch-european-early warning-system-wheat-rust, accessed on 12 October 2022), up to four races have been found for this pathogen. In addition, while some known major genes conferring resistance on *Warrior (-)* races, such as *Yr5, Yr8, Yr110, Yr15, and Yr27*, are still effective in the European wheat population, other genes, such as *Yr1, Yr2, Yr3, Yr4, Yr6, Yr7, Yr9, Yr17, Yr25, Yr32, YrSp,* and *YrAvS*, have been overcome by this pathotype. Although a steady increase in seedling resistance of European winter wheat materials has been reported [21], breeding progress in the seedling resistance of these materials could not be attributed to single resistance genes (R genes), most likely because of the multitude of pathogen virulence and R genes affecting the susceptibility of a single cultivar [21].

To find the sources of effective resistance alleles and associated QTLs for use in breeding programs in this panel, SNPs identified by the GWAS were compared with known yellow rust resistance genes and QTLs reported in previous studies based on available physical locations and genetic maps of flanking markers (Appendix A). Based on QTLs obtained from the average of infection types from three replicate trials for *Warrior (-) FS 53/20*, physical overlaps were observed for the associated marker *Tdurum_contig13879_352* with eight QTLs: *QYrex.wgp-1BL_Express* [22], *QYr.sun-1B_Kukri* [23], *QYr.sun-1B_CPI133872* [24], *QYr.sun-1B_Wollaroi* [25], *QYr.ucw-1B* [8], *QYr.jic-1B_Guardian* [26], *QYr.tam-1B_Quaiu* [27], *QYr.cim-1BL_Francolin* [28], and *Yr29/Lr46* on chromosome 1B, as reported by [29]. The QTLs *Qyr.yellow-1BL* and *QYr-1BL.2*, previously described by Pradhan et al. [30] and Zhang et al. [31], respectively, coincided with the genomic region containing *Yr29*. *Yr29* shares the same features as *Yr18*, which is associated with a weak compatible infection type at the seedling stage. However, seedlings with *Yr18* exposed to low temperatures (e.g., 6–9 °C) show typical partial resistance with longer latency, smaller lesions, and less sporulation. This phenomenon is likely to be important for slowing winter and early spring epidemics in winter wheat. These genes all confer non race-specific and presumably durable resistance and are effective against multiple pathogens, making them very important resistance genes in wheat [12]. However, further studies are needed to confirm whether the QTL identified on chromosome 1B is allelic to or distinct from *Yr29/Lr46*.

The SNPs on chromosomes 2A, 5B, and 7A are physically located in the same genomic region as five previously published QTLs: *QYr.inra_2AL.2_Camp Remy* [32]; *QYr.sun-5B_Janz* [23]; *QYr.caas-5BL.3_SHA3/CBRD* [33]; and *QYr.cim-7AS_Avocet* [34]. Finally, for *Warrior (-) FS 53/20*, we discovered a new QTL for yellow rust resistance associated with the marker *Kukri_c5357_323* at the physical position of 637.621 Mb, distal to the previously reported QTL on chromosome 1B. For *Warrior (-) G 23/19*, two overlapping QTLs, *QYr.sun-5B_Wollaroi* [27] and *QYr.ui-5B_IDO444* [35], were found in the examined physical region of the associated marker *Excalibur_c17489_804* on chromosome 5B.

We compared the results of our previous study [16], using the same association panel at the adult plant stage in the field (excluding the line TS 175), with the results of the present study, which were obtained from seedling stage evaluation in the greenhouse. We found that the breeding line TS185 was the only variety resistant to stripe rust at both growth stages. The correlation coefficients between the seedling and adult plant stages were low (−0.01 to 0.25) because most lines were susceptible at the seedling stage but resistant in the field trials. Such a low correlation (0.19 and 0.46) was also found in the studies by Yao et al. [5] and Rollar et al. [20]. While several studies [5,15,19,20] have identified similar QTLs controlling resistance to yellow rust at both seedling and adult stages, suggesting the presence of resistance genes for all stages, others [36,37], including our study, have not found common QTLs or representative SNP loci between these two developmental stages. This divergence may be because many other loci contribute more to resistance in the fields in response to a population of pathotypes, while in seedling tests there is only a limited response to a few races. This discrepancy could also be due to the interaction between host and pathogen at the seedling stage in the greenhouse and at the adult plant stage in the field. The seedling plants responded to two actual isolates of the dominant *Warrior (-)* race in the greenhouse. However, there might be a small difference in the virulence spectrum of this race in the field that causes different behaviour of the lines. In addition, seedling tests are usually conducted to postulate major genes and identify resources with new genes that are then used in systematic breeding. They capture gene-by-gene relationships and overlook quantitative effects, whereas field tests capture both types of resistance, all-stage and adult plant resistance (APR) genes. In this case, some breeding approaches, such as genomic prediction models, need to be trained on field data to be used as reliable selection tools.

In summary, our study identified resistant genotypes and the potential source of effective resistance alleles and associated QTLs that could be used to improve yellow rust resistance levels in the current Northern and Central European winter wheat breeding materials.

## 4. Materials and Methods

### 4.1. Seedling Stage Yellow Rust Assessment

We selected a population of 229 winter wheat cultivars and breeding lines for the GWAS to capture broad genetic variability in Europe (Appendix A). Using breeder knowledge and the coefficient of determination algorithm [38], we collected genotypes from Germany, Austria, Norway, Sweden, Denmark, Poland, and Switzerland comprising 157, 50, 14, 4, 3, 1, and 1 genotype(s), respectively [16].

Seedling screening against *Warrior (-)* yellow rust was performed under controlled conditions in the greenhouse at the Bayerische Landesanstalt für Landwirtschaft, Institut für Pflanzenbau und Pflanzenzüchtung, Freising, Germany. The association mapping panel was screened for seedling resistance to two *PstS10* pathotypes, *Warrior (-) FS 53/20* and *Warrior (-) G 23/19*, kindly provided by Dr. Kerstin Flath, JKI, Germany. Pathotypes were selected based on their virulence and prevalence according to the RustWatch project (https://ec.europa.eu/eip/agriculture/en/find-connect/projects/rustwatch-european-early warning-system-wheat-rust, accessed on 12 October 2022) for wheat. A total of 8 to 12 seeds of 5 genotypes were planted in 14 cm × 14 cm × 9 cm pots. The mixture in the pots consisted of soil, compost, and sand in a 1:1:1 ratio. Seedlings were grown in a growth chamber at 14–16 °C night/day temperature for two weeks. Plants were inoculated at the two-leaf stage by preparing the inoculum immediately before use and suspending urediospores in a solution of water and ~1ppm surfactant (Tween20) or 0.1% agarose. The solution was sprayed onto the seedlings, which were then incubated for 24 h in the dark and in moist plastic cages with a relative humidity of close to 100% at 14–16 °C. Plants were then maintained at a similar temperature of 14–16 °C, with 16 h of light at 15 lux, and 8 h of darkness until disease development. Yellow rust was evaluated 18–21 days after inoculation at the time of clearly visible disease symptoms on the leaves of the susceptible standard cultivar “Akteur” using a scale of 1–9 infection types (ITs) [39]. According to this scale, infection types 1–9 are defined as follows: 1: minor chlorotic and necrotic flecks, 2: chlorotic and necrotic flecks without sporulation, 3–4: chlorotic and necrotic areas with limited sporulation, 5–6: chlorotic and necrotic areas with moderate sporulation, 7: abundant sporulation with moderate chlorosis, and 8–9: abundant and dense sporulation without notable chlorosis and necrosis. This scale is commonly used as a uniform system for recording and processing stripe rust research data. We chose this method because of its feasibility and accuracy in evaluating seedling plants in the greenhouse in response to yellow rust. Seedling ITs were rated 0–4 (R) if resistant, 5–6 (MR) if moderately resistant, and 7–9 (MS) if moderately to highly susceptible [15]. Evaluation of lines in response to each pathotype was repeated three times from October 2021 to April 2022, and the mean value recorded for IT scores was used for analyses.

### 4.2. Statistical Analysis

Descriptive statistics, analysis of variance (ANOVA), and correlation analysis were performed using “PROC GLM” and “PROC CORR” of the SAS statistical package v.9.4 (SAS Institute, Cary, NC, USA).

### 4.3. SNP Genotyping, Population Structure, and LD Analyses

Genomic DNA extraction, SNP genotyping, population structure, and LD analyses were performed as described in Shahinnia et al. [16]. Briefly, genotyping was performed using the 25 K Infinium iSelect array (TraitGenetics, Seeland OT Gatersleben, Germany), and monomorphic SNPs and SNPs with more than 10% missing values and a minor allele frequency of less than 5% were excluded from further analysis using the package “synbreed” in R [40]. The physical location of markers was determined by BLAST using the published Chinese Spring genome sequence (IWGSC RefSeq v1.0). For population structure, LD, and genetic association analysis, 8812 informative and polymorphic SNP markers (Appendix A) with an average minor allele frequency of 0.26 were used. Linkage disequilibrium (LD) between markers was estimated for the association mapping panel using observed and expected allele frequencies in TASSEL 5.2.78 [41], as reported in Shahinnia et al. [16]. The genetic structure of the population was determined using the Bayesian clustering program STRUCTURE, and the results of STRUCTURE were analysed in STRUCTURE HARVESTER [42,43]. This population was divided 2 subpopulations (Appendix A) with 92 Austrian breeding lines and cultivars (group 1) separated from the other 137 genotypes (group 2) from Germany, Norway, Sweden, Denmark, Poland, and Switzerland, previously [16].

### 4.4. Association Mapping

Marker–trait association analysis was carried out using a mixed linear model that accounted for population structure (**Q**) and kinship matrix (**K**). The model can be described as y=Xβ+Zu+e**,** where **y** is the vector of observations, **β** is a vector containing fixed effects for genetic markers and population structure (**Q**), **u** is a vector of random additive genetic effects from multiple background QTLs with **u** ~ N(0, $\sigma_{G}^{2}$**K**), **X** and **Z** are the known design matrices, and **e** is a vector of random residuals with **e** ~ N(0, $\sigma_{G}^{2}$**I**). To provide adjusted *P*-values, the false discovery rate (FDR) was calculated using a threshold of <5% with the “q-value” package in R. To evaluate the performance of the models and appropriate thresholds, QQ plots were drawn in TASSEL. Associations of SNP markers with yellow rust severity were represented by drawing Manhattan plots. The physical location of significantly associated markers was compared with previously published *Yr* genes and QTLs using *The Catalogue of Gene Symbols for Wheat* (https://wheat.pw.usda.gov/GG3/WGC, accessed on 12 October 2022) [44] and an integrated map for chromosomal locations of loci associated with responses to *Pst* from Bulli et al. [13] and Maccaferri et al. [8].

## Figures and Tables

**Figure 1 plants-12-00420-f001:**
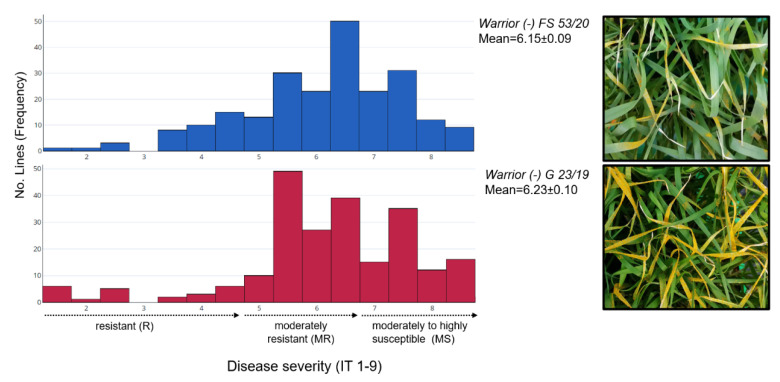
Frequency distribution of yellow rust disease severity (based on infection type) for association panel evaluated with *Pst* pathotypes *Warrior (-) FS 53/20* and *Warrior (-) G 23/19*. Typical uredinia of yellow rust caused by each *Pst* pathotype are shown on the right.

**Figure 2 plants-12-00420-f002:**
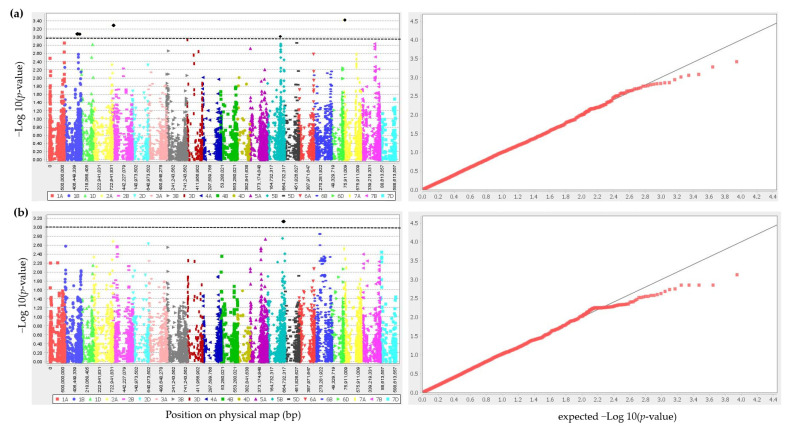
Manhattan plots showing the association of SNPs in the 229 genotypes (**left**) and QQ plot comparing the performance of the mixed linear model (**right**) used in the genome-wide association study for yellow rust resistance at the seedling stage for *Pst* pathotypes (**a**) *Warrior (-) FS 53/20* and (**b**) *Warrior (-) G 23/19*. The black dots and horizontal lines represent the genome-wide significant SNPs and thresholds in Manhattan plots.

**Table 1 plants-12-00420-t001:** SNP markers associated with yellow rust severity (based on IT) for the means of each *Pst* pathotype. The marker alleles and IT associated with increased resistance are bolded. IT indicates the mean infection type corresponding to the resistant vs. susceptible allele.

Pathotype	Marker	Chromosome	Position (bp)	*R^2^*	SNPs	Number of Accessions with Resistant Allele	Number of Accessions with Susceptible Allele	Number of Accessions with Missing Alleles	IT	Effect	*p*-Value
*Warrior (-) FS 53/20*											
	*Kukri_c5357_323*	1B	637,621,766	0.051	**G**/T	18	210	1	**5.00**/6.24	1.20	0.0008
	*Tdurum_contig13879_352*	1B	680,162,719	0.051	**A**/G	193	34	2	**6.00**/6.91	0.91	0.0009
	*BS00098033_51*	2A	778,724,092	0.055	**C**/T	19	205	5	**5.02**/6.26	1.16	0.0005
	*CAP12_c703_150*	5B	550,377,847	0.053	**C**/T	95	126	8	**5.89**/6.38	0.76	0.0010
	*BS00062869_51*	7A	17,454,693	0.061	**A**/G	21	205	3	**5.06**/6.27	1.14	0.0004
*Warrior (-) G 23/19*											
	*Excalibur_c17489_804*	5B	670,829,789	0.052	C/**T**	104	120	5	6.48/**6.01**	0.81	0.0008

**Table 2 plants-12-00420-t002:** The resistant lines contained at least 3 alleles (shown by X) of 6 significant SNP sites identified by GWAS at the seedling stage for the IT means of the *Pst* pathotype.

SNP/IT	*Kukri_c5357_323*	*Tdurum_contig13879_352*	*BS00098033_51*	*CAP12_c703_150*	*BS00062869_51*	*Excalibur_c17489_804*	IT Against *Warrior (-)FS 53/20*	IT Against *Warrior (-) G23/19*
Line/Allele	G	A	C	C	A	T		
TS049		X		X		X	3.67	2.33
TS111	X	X	X	X	X		1.67	4.33
TS185		X	X	X	X	X	2.33	2.67
TS229		X	X	X	X		2.01	2.33

## Data Availability

The original contributions presented in the study are included in the article/Appendix A. Further inquiries can be directed to the corresponding author.

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
