# Peer review of "Genetic Basis of Resistance to Warrior (-) Yellow Rust Race at the Seedling Stage in Current Central and Northern European Winter Wheat Germplasm"

_plants, 2023, doi:10.3390/plants12030420_

Round 1

Reviewer 1 Report (New Reviewer)

The authors identified a wide variation in yellow rust disease severity using a population of 229 winter wheat cultivars and breeding lines for GWAS. Four breeding lines and one cultivar from Germany were found to be resistant to Warrior (-) FS 53/20 and 15 Warrior (-) G 23/19. GWAS identified five significant SNPs associated with yellow rust. For Warrior (-) FS 53/20, they discovered a new QTL for yellow rust resistance associated with the marker Kukri_c5357_323 on chromosome 1B. The resistant alleles G and T at the marker loci Kukri_c5357_323 on chromosome 1B and Excalibur_c17489_804 on chromosome 5B showed the largest effects (1.21 and 0.81, respectively) on the severity of Warrior (-) FS 53/20 and Warrior (-) G 23/19. Among 144 putative genes within the flanking sequence of the 6 significant SNPs detected by GWAS, the function of the best candidate genes was determined as protein kinase activity and oxidoreductase activity. However, from the view of a GWAS analysis article, I think the results of this manuscript are too preliminary.

Major concerns:

1.     Three inoculation replicates were described in lines 291-292, and the author directly took the mean value for association analysis. In my opinion, association analysis should be conducted separately for each repeat, and the quality of the repeats should be assessed. Then make sure to use the mean value or Blup for association analysis.

2.     The quality of the Manhattan plots is very poor, with no clear peak interval.

3.     Line 151, the article does not specify the basis for searching genes in the '2Mb" range.

Minor concerns:

1.      Add the infection types corresponding to alleles in Table 1 (Phenotype corresponding to genotypes of significant SNPs).

2.      Whether these resistant varieties (TS049, TS111, TS185, and TS229) contain elite alleles of significant SNP sites from Table 1 or not.

3.      To provide the expression data of candidate genes (mentioned in lines 154-156) before and after inoculation.

Author Response

Thank you for your valuable suggestions that helped improve our manuscript. Here is the response to your suggestions:

Major concerns:

  1. Three inoculation replicates were described in lines 291-292, and the author directly took the mean value for association analysis. In my opinion, association analysis should be conducted separately for each repeat, and the quality of the repeats should be assessed. Then make sure to use the mean value or Blup for association analysis.

We understand the reviewer's concerns regarding the use of the mean values for the association analysis. We would like to clarify that although the three independent experiments were conducted under control conditions in the greenhouse, it is recommended to use the average of the data sets due to possible uncontrolled environmental changes. Based on our previous observations, the repeatability of greenhouse experiments for yellow rust is not as high as other fungi diseases such as leaf rust in wheat. This is highly dependent on environmental conditions such as temperature, humidity, and day length, so that conducting GWAS based on a single replication is not reliable enough for reporting GWAS results. In addition, we found that the correlations (r2= 0.39 to 0.54) between replications for the Warrior (-) FS 53/20 were smaller than the correlations (r2= 0.78 to 0.82) for Warrior (-) G 23/19 in our study. Therefore, to obtain a consistent analysis, we used the average of the data sets.

  1. The quality of the Manhattan plots is very poor, with no clear peak interval.

We improved the figure of the Manhattan plots. We hope that it is now satisfying.

  1. Line 151, the article does not specify the basis for searching genes in the '2Mb" range.

One of the applications of candidate gene identification is the use of this information in map-based cloning of genes, e.g., in biparental populations in further studies. In this case, it is necessary to perform fine mapping for the flanking markers of the QTL to achieve at least a genetic distance of less than 1cM. Although the genetic distance varies throughout the genome, it could be around 2Mb (1 Mb upstream and downstream) from the peak of each targeted marker in the wheat genome. This distance has also been used in other studies to identify candidate genes (Muqaddasi et al. Scientific Reports (2020) 10:12541; Shahinnia et al. TAG (2022) 135: 3583–3595.

Minor concerns:

  1. Add the infection types corresponding to alleles in Table 1 (Phenotype corresponding to genotypes of significant SNPs).

We have indicated in this table the mean infection type (IT) corresponding to the resistant allele. Due to the oligogenic to multigenic inheritance of disease resistance, these values are around the mean plus/minus the effect reported in the table.

  1. Whether these resistant varieties (TS049, TS111, TS185, and TS229) contain elite alleles of significant SNP sites from Table 1 or not.

These lines contain at least 3 alleles (showing by X) of 6 significant SNP sites from Table 1 as showing in the below table:

Line/Allele

G

A

C

C

A

T

TS49

X

X

X

TS111

X

X

X

X

X

TS185

X

X

X

X

X

TS229

X

X

X

X

  1. To provide the expression data of candidate genes (mentioned in lines 154-156) before and after inoculation.

Expression analysis of candidate genes was not within the scope of the current study. Unfortunately, we do not have expression data from the evaluation of seedling plants before and after inoculation to present.

Reviewer 2 Report (New Reviewer)

The manuscript entitled “Genetic basis of resistance to Warrior(-) Yellow Rust race at the seedling stage in current central and northern European Winter wheat Germplasm” by Shahinnia et. al. aimed to study the association between genetic variability and seedling plant response to a dominating Warrior(-) race of yellow rust in Northern and Central European germplasm by using 229 winter wheat cultivars. 

The author found five significant SNPs associated with yellow rust on chromosomes 1B, 2A, 5B and 7A. Among 144 putative genetic variability, the best candidate genes were determined as protein kinase activity and oxidoreductase activity. This study provides the basis for knowledge-based resistance breeding in the impact of the Warrior race on wheat production. 

Overall, the authors presented data analysis from GWAS. The method used in the study is thorough. Conclusions are appropriate, and supported by the data. Statistical analysis is provided within the manuscript. The whole study is sound, and I recommend accepting it. 

Author Response

Thank you for your positive feedback and valuable evaluations of our manuscript.

Reviewer 3 Report (New Reviewer)

The manuscript presents research on the genetic basis of resistance to Warrior yellow rust race at the seedling stage. The authors chose the correct approach to identify putative new loci contributing to the resistance to this important pathogen - by applying GWAS on 229 accessions of diverse origin and with a varying degree of susceptibility/tolerance to the pathogen. Upon identification of the QTLs they made expansive searches up to 2 Mbases upstream and downstream of each locus for genes with known functions contributing to pathogen response.

The manuscript is well written, with clear results and adequate discussion.

The only minor comment from me is that authors could have used "QTL" and "QTLs" as appropriate to denote whether single or multiple loci are envisaged in each particular case as this facilitates the reading and makes understanding of the results and the discussion easier.

Author Response

Thank you for your positive feed back and valuable suggestion that helped improve our manuscript. Here is the response to your suggestion:

The only minor comment from me is that authors could have used "QTL" and "QTLs" as appropriate to denote whether single or multiple loci are envisaged in each particular case as this facilitates the reading and makes understanding of the results and the discussion easier.

We have done this correction in the text as suggested by the reviewer.

Round 2

Reviewer 1 Report (New Reviewer)

1.      I don't quite agree with the author's answer to my question 1. The authors emphasize the credibility of their SNPs in the abstract of their own another paper: Genome-wide association study for adult plant stripe rust resistance identified 12 SNP markers on six wheat chromosomes which showed consistent effects over several testing environments. It can be seen that the repeatability of different environmental points is very important to accurately determine whether SNP is associated with traits. If the environmental factors have a great influence, it is not advisable to choose the average value. The number of biological repeats should be increased, and samples that can be replicated should be selected for analysis, rather than average values.

2.      I am also not satisfied with the author's answer to my second question. I don't think the Manhattan plot has been improved.

3.      The authors answered: Although the genetic distance varies throughout the genome, it could be around 2Mb (1 Mb upstream and downstream) from the peak of each targeted marker in the wheat genome. This distance has also been used in other studies to identify candidate genes (Muqaddasi et al. Scientific Reports (2020) 10:12541; Shahinnia et al. TAG (2022) 135: 3583–3595. I think the author lacks sufficient evidence for this judgment. The size of this distance depends on the population size, marker density and many other factors. The author cited a paper published by his own laboratory and another one as evidence. However, I don't think these two articles are enough to support that 2Mb is reasonable in this job.

Author Response

  1. The GWAS results for three replicates with respect to each isolate are shown in Table S2 and in the text (l. 144-152). As we mentioned earlier, we recommend to use the average of the data sets because of the instability of the reaction, although the three independent experiments were conducted under control conditions in the greenhouse. Based on our observations to date, the repeatability of greenhouse trials for yellow rust is not as high as for other fungal diseases such as leaf rust in wheat. This is highly dependent on environmental conditions such as temperature, humidity, and day length, so conducting GWAS based on a single replication is not reliable enough to report GWAS results. This situation is completely different under uncontrolled environmental conditions in the field and is not comparable to what we previously published in TAG (Shahinnia et al. 2022). In addition, we found that the correlations (r2= 0.39 to 0.54) between replications for Warrior (-) FS 53/20 were smaller than the correlations (r2= 0.78 to 0.82) for Warrior (-) G 23/19 in our study. This information was added to the text (lines 114-116). To maintain a consistent analysis, we used the average of the data sets to present the results in the main text and included the GWAS for replications in the supplemental materials.

  1. Unfortunately, this is the maximum resolution we can achieve with the software. We have enlarged the image and shown the significant SNPs and their thereshold lines in black to clearly highlight the most important results there.

  2. We have been asked by another reviewer to remove the results and discussion related to the candidate genes in the current version.

This manuscript is a resubmission of an earlier submission. The following is a list of the peer review reports and author responses from that submission.

Round 1

Author Response

Response to Reviewer 1 Comments

We would like to thank you for your valuable suggestions that helped improve our manuscript. 

We have answered all the questions you raised point by point as follows:

Point 1. I have doubts in performing GWAS for the data related to disease resistance where quantitative score could not be obtained. Such type of data becomes a categorical data and GWAS results needs to be validated in bi-parental population.

Disease responses to cereal rusts can be evaluated by either qualitative or quantitative means, or a combination of both (McIntosh et al., Wheat Rusts: An Atlas of Resistance Genes. Australasian Plant Pathology 25, 70 (1996). https://doi.org/10.1007/BF03214019). This is based on the knowledge that each resistance gene produces characteristic responses related to infection type, infection site variability on a single leaf, and environmental variation. Infection type (1-9) is only one of the four methods of disease evaluation proposed by Roelfs (1985), the others being the number of uredia per unit inoculum (i.e., host susceptibility), length of latency (i.e., time for pustule formation), and duration of sporulation. Descriptions of infection types still in common use are based on the original scales proposed by Stakman et al. (1962) for stem rust and by Gassner and Straib (1932) for yellow rust. In research on yellow rust, the 0-9 scale of McNeal et al. (1971) is commonly used as a uniform system for recording and processing stripe rust research data (McIntosh et al. 1996). We chose this method because of its feasibility and accuracy in evaluating seedling plants in the greenhouse in response to yellow rust. This information was added in the text (Lines 296-299). Moreover, the histograms in our manuscript (Figure 1) show a quantitative distribution of our data, and this type of GWAS has been published several times, e.g., by Tehseen et al. (2021), Mahmood et al. (2021), and Gao et al. (2016), who were cited in our manuscript.

We admire that it is always useful to validate GWAS results using a biparental population. However, this was beyond our capabilities as it could take a lot of time and lead to a stage where we are confronted with a new Warrior (-) yellow rust race. For this reason, we would prefer to share our current results with the wheat research and breeding community to develop and use resistant varieties using breeding methods for ecologically control of this disease in Europe.

Point 2. It is difficult for me to believe on the DNA regions which has been found to be associated to rust resistance. Since identification of resistance also becomes a rare phenomenon and few lines are always found in the germplasm showing clean resistance. Any rare SNPs or SNPs with lower MAF will show association.

As we have shown in Table S3, Results and Discussion, for the interval of 5 of the 6 major loci identified by GWAS in our manuscript, at least 13 QTL and one major gene have been previously reported in the context of stripe rust resistance in wheat. In addition, before performing the GWAS analysis, we had excluded the monomorphic SNPs and SNPs with more than 10% missing values and minor allele frequency of less than 5% from further analysis (this information was added to the text, lines 312-314). In this way, we are confident that all rare SNPs with low frequency have been appropriately filtered out of the GWAS analysis. 

Point 3. Even threshold to say significant is very low (whether it is bonferreni threshold or any other).

To obtain adjusted P values, the false discovery rate (FDR) was calculated, resulting in very stringent thresholds with P values less than 0.0001. This analysis is highly accepted and reputed across the literature for performing GWAS. This could very well be explained by the small number of SNPs that were finally considered as the significant SNPs associated with stripe rust resistance. It is worthy to mention that we have previously published our results from evaluation of same association panel at the adult plant stage using the FDR method to obtain the significant thresholds (Shahinnia et al., 2022, TAG).

Point 4. The phenotypic contribution is SNP is also very low, which may not be accepted at least in case of resistance loci.

Such QTL with low effect for resistance to yellow rust at the seedling stage have also been identified in the American elite spring wheat lines [19], in the German MAGIC winter wheat population [20], and in a Chinese wheat landrace association panel [5]. The genetic control of this disease is quantitative, with a relatively large number of genes with moderate effect. Therefore, the phenotypic contribution of associated SNPs is small. More importantly, about 10% of the lines showed a high level of resistance, which could be caused by major resistance genes to both pathotypes or by a combination of a few resistance genes. Here, we would like to refer to one of our previous studies conducted by Albrecht and Mohler (personal communication, unpublished) in which using a couple of diagnostic markers obtained from cloning of Yr15 gene (Klymiuk et al. 2018, Nature communication), as one of the most effective resistant genes to Warrior (-), the presence of this gene were confirmed only in 2 lines out of a diversity panel of nearly 1000 cultivars and breeding lines belonging to the current European winter wheat germplasm including our association panel. This means that although the single major resistance genes could be existing in this population, but they remained undetected in GWAS due to their low allele frequencies. We have explained these issues in our Discussion in details (Lines 170-193).

Reviewer 2 Report

It is crucial to understand the genetic variation controlling resistance to yellow rust (known as stripe rust, Puccinia striiformis) in wheat population. Such understanding can help wheat resistance breeding by combining not only major and minor resistance genes but also race-specific and -nonspecific genes. This research investigated wheat genetic variability and seedling plant response to a dominated Warrior (-) race of yellow rust in Northern and Central European germplasm by performing GWAS on a population of 229 winter wheat cultivars and breeding lines. The authors found four breeding lines showed resistant to Warrior (-) FS 53/20 and Warrior (-) G 23/19, identified five significant SNPs and a new QTL associated with yellow rust. The derived results can be used in wheat resistance breeding against yellow rust. In this study, the authors used standard GWAS method and commonly used software for statistical analysis.

One of my major concerns is whether the population size (229 winter wheat cultivars) is large enough to conduct such GWAS research. Also, I would request the authors to justify whether the disease score approach is an accurate, quantitative method for measure disease phenotypic traits in a GWAS or QTL mapping research. Are there any better solutions for quantitative phenotypic data collection?

For the results presentation, the box plots of disease severity for Warrior (-) FS 53/20 and Warrior (-) G 23/19 in Figure 1 doesn’t speak to me. I would merge the yellow rust phenotypes (top) with histogram plots in Figure. 2, which I think can better capture the distribution of disease phenotypes. In this study, the identified gene candidates and associated markers are important for wheat resistance breeding. I would provide more analysis results about such gene candidates and associated markers in Result section. For example, some comparison of results derived from this study and previous studies in Discussion section.

Author Response

Response to Reviewer 2 Comments

We would like to thank you for your valuable suggestions that helped improve our manuscript. 

We have answered all the questions you raised point by point as follows:

Point 1. One of my major concerns is whether the population size (229 winter wheat cultivars) is large enough to conduct such GWAS research.

The size of our population is large enough for GWAS. The results of our recent GWAS on yellow rust resistance at the adult plant stage using the same population was published in TAG (Shahinnia et al. 2022. https://doi.org/10.1007/s00122–022–04202–z.).

Point 2. I would request the authors to justify whether the disease score approach is an accurate, quantitative method for measure disease phenotypic traits in a GWAS or QTL mapping research. Are there any better solutions for quantitative phenotypic data collection?

Disease responses to cereal rusts can be evaluated by either qualitative or quantitative means, or a combination of both (McIntosh et al., Wheat Rusts: An Atlas of Resistance Genes. Australasian Plant Pathology 25, 70 (1996). https://doi.org/10.1007/BF03214019). This is based on the knowledge that each resistance gene produces characteristic responses related to infection type, infection site variability on a single leaf, and environmental variation. Infection type (1-9) is only one of the four methods of disease evaluation proposed by Roelfs (1985), the others being the number of uredia per unit inoculum (i.e., host susceptibility), length of latency (i.e., time for pustule formation), and duration of sporulation. Descriptions of infection types still in common use are based on the original scales proposed by Stakman et al. (1962) for stem rust and by Gassner and Straib (1932) for yellow rust. In research on yellow rust, the 0-9 scale of McNeal et al. (1971) is commonly used as a uniform system for recording and processing stripe rust research data (McIntosh et al. 1996). We chose this method because of its feasibility and accuracy in evaluating seedling plants in the greenhouse in response to yellow rust. This information was added in the text (Lines 296-299).

Point 3. For the results presentation, the box plots of disease severity for Warrior (-) FS 53/20 and Warrior (-) G 23/19 in Figure 1 doesn’t speak to me. I would merge the yellow rust phenotypes (top) with histogram plots in Figure. 2, which I think can better capture the distribution of disease phenotypes.

In the revised version, we have deleted the box plots and only kept the histogram plots together with the uredinia of yellow rust caused by each Pst pathotype presented as Figure 1.

Point 4. In this study, the identified gene candidates and associated markers are important for wheat resistance breeding. I would provide more analysis results about such gene candidates and associated markers in Result section. For example, some comparison of results derived from this study and previous studies in Discussion section.

The additional information related to the identification of previously published Yr genes and QTLs was added to the Results (lines 146-154) and Table S2, highlighted in red, and in the text (lines 359-361). We also discussed this information earlier (lines 194-239).

Author Response

Response to Reviewer 3 Comments

We would like to thank you for your positive feed backs and valuable suggestions that helped improve our manuscript. 

We have answered the question you raised as follows:

Point 1. Citations before 2005 may be avoided.

We have tried to use the original citations as the first published study on the related topic. The citations prior to 2005 that we have used here must be retained because they either refer to a method or are among the rare publications in the field.

Round 2

Reviewer 1 Report

Dear Authors

I appreciate your previous publications suggested by you for raising your points. However, under point number 4, you have also mentioned that major genes present in the resistant germplasm lines are usually not captured in GWAS due to low allelic frequency.  Besides, very few lines in the association panel show resistance, and resistant alleles in these lines mostly get filtered in MAF filtration. GWAS in this publication could only retrieve the minor alleles whose role in resistance is not confirmed and will be very difficult to confirm until unless these are combined with major QTL alleles. 

This publication has one achievement which is the identification of donor lines showing resistance to the Warrior race at the seedling stage. However the major emphasis of this paper is on GWAS. My suggestion is to capture major QTL alleles for resistance in highly resistant lines using bi-parental population, which i am sure, could not be captured due to the low frequency of these lines in germplasm. 

The alleles captured in this publication do not find any direct value in breeding for yellow rust resistance at the moment. 

With all these observations, I may not recommend it for publication.